# Structural and Social Determinants of Health Factors Associated with County-Level Variation in Non-Adherence to Antihypertensive Medication Treatment

**DOI:** 10.3390/ijerph17186684

**Published:** 2020-09-14

**Authors:** Macarius M. Donneyong, Teng-Jen Chang, John W. Jackson, Michael A. Langston, Paul D. Juarez, Shawnita Sealy-Jefferson, Bo Lu, Wansoo Im, R. Burciaga Valdez, Baldwin M. Way, Cynthia Colen, Michael A. Fischer, Pamela Salsberry, John F.P. Bridges, Darryl B. Hood

**Affiliations:** 1College of Pharmacy, Ohio State University, Columbus, OH 43210, USA; chang.1065@osu.edu; 2Departments of Epidemiology and Mental Health, Johns Hopkins Bloomberg School of Public Health, Baltimore, MD 21205, USA; john.jackson@jhu.edu; 3Department of Electrical Engineering and Computer Science, University of Tennessee, Knoxville, TN 37996, USA; langston@tennessee.edu; 4Department of Family and Community Medicine, Meharry Medical College, Nashville, TN 37208, USA; pjuarez@mmc.edu (P.D.J.); wim@mmc.edu (W.I.); 5College of Public Health, Ohio State University, Columbus, OH 43210, USA; sealy-jefferson.1@osu.edu (S.S.-J.); lu.232@osu.edu (B.L.); colen.3@osu.edu (C.C.); salsberry.1@osu.edu (P.S.); hood.188@osu.edu (D.B.H.); 6Family & Community Medicine, University of New Mexico, Albuquerque, NM 87131, USA; rovaldez@unm.edu; 7Department of Psychology, Ohio State University, Columbus, OH 43210, USA; way.37@osu.edu; 8Division of Pharmacoepidemiology and Pharmacoeconomics, Brigham & Women’s Hospital, Boston, MA 02115, USA; MFISCHER@BWH.HARVARD.EDU; 9Department of Biomedical Informatics, Ohio State University, Columbus, OH 43210, USA; john.bridges@osumc.edu

**Keywords:** adherence, antihypertensive medications, hypertension, social determinants of health, county health rankings, CDC Atlas

## Abstract

Background: Non-adherence to antihypertensive medication treatment (AHM) is a complex health behavior with determinants that extend beyond the individual patient. The structural and social determinants of health (SDH) that predispose populations to ill health and unhealthy behaviors could be potential barriers to long-term adherence to AHM. However, the role of SDH in AHM non-adherence has been understudied. Therefore, we aimed to define and identify the SDH factors associated with non-adherence to AHM and to quantify the variation in county-level non-adherence to AHM explained by these factors. Methods: Two cross-sectional datasets, the Centers for Disease Control and Prevention (CDC) Atlas of Heart Disease and Stroke (2014–2016 cycle) and the 2016 County Health Rankings (CHR), were linked to create an analytic dataset. Contextual SDH variables were extracted from the CDC-CHR linked dataset. County-level prevalence of AHM non-adherence, based on Medicare fee-for-service beneficiaries’ claims data, was extracted from the CDC Atlas dataset. The CDC measured AHM non-adherence as the proportion of days covered (PDC) with AHM during a 365 day period for Medicare Part D beneficiaries and aggregated these measures at the county level. We applied confirmatory factor analysis (CFA) to identify the constructs of social determinants of AHM non-adherence. AHM non-adherence variation and its social determinants were measured with structural equation models. Results: Among 3000 counties in the U.S., the weighted mean prevalence of AHM non-adherence (PDC < 80%) in 2015 was 25.0%, with a standard deviation (SD) of 18.8%. AHM non-adherence was directly associated with poverty/food insecurity (β = 0.31, *P*-value < 0.001) and weak social supports (β = 0.27, *P*-value < 0.001), but inversely with healthy built environment (β = −0.10, *P*-value = 0.02). These three constructs explained one-third (*R^2^* = 30.0%) of the variation in county-level AHM non-adherence. Conclusion: AHM non-adherence varies by geographical location, one-third of which is explained by contextual SDH factors including poverty/food insecurity, weak social supports and healthy built environments.

## 1. Introduction

Nearly half of hypertension patients [1,2] are non-adherent to antihypertensive medication treatment (AHM). Non-adherence to AHM is associated with a 27% higher risk of stroke [3] and approximately 56% higher risk of cardiovascular diseases (CVDs) [4]. The reasons for suboptimal AHM adherence are still not well understood, despite extensive research and innovative strategies to improve this complex health behavior [5]. Research on the determinants of non-adherence to AHM has focused on individual patient and provider characteristics. However, these factors do not fully explain variations in medication adherence [6,7,8].

The structural and social determinants of health (SDH) that predispose populations to ill health and unhealthy behaviors could be potential barriers to long-term adherence to AHM. Several studies have reported social support, food insecurity, poverty and lack of transportation as individual-level SDH that are associated with medication non-adherence [9,10,11,12,13,14]. However, the relationships between AHM non-adherence and contextual SDH, i.e., SDH factors measured at the community level, have not yet been well investigated [15]. Emerging, but limited, data suggests that residential location and some contextual SDH are potential predictors of medication non-adherence. Nationally representative data from Medicare and commercial health plan beneficiaries in the U.S. showed that adherence to AHM as well as antidiabetic and antilipid medications varied by geographic regions [16]. A few studies, all focusing on oral antidiabetic medications, reported that medication adherence is inversely associated with neighborhood social affluence, residential stability, socioeconomic advantage and safety [17,18,19,20].

While evidence for a link between contextual SDH factors and medication non-adherence is beginning to grow, this area of research is stymied by lack of access to data and methodological limitations. Administrative health care claims databases are routinely used to measure non-adherence based on prescription fill data. However, these databases do not capture data on the social and structural conditions in which patients are living. There are a growing number of databases such as the Public Health Exposome [21], the County Health Rankings (CHR), and the Centers for Disease Control and Prevention (CDC) Atlas for Heart Disease and Stroke with information on contextual SDH factors. However, linking these databases with claims databases remains a challenge because of patient data privacy issues. Another challenge is the lack of a standard methodological framework defining the relationships between contextual SDH factors and medication non-adherence.

The primary objective of this analysis was to define and identify the social determinants of non-adherence to AHM and to quantify the variation in county-level non-adherence to AHM explained by these factors. The relationships between these constructs and AHM non-adherence were then tested with structural equation modeling.

## 2. Materials and Methods

We leveraged data on county-level prevalence of AHM non-adherence from the CDC Atlas of Heart Diseases and Stroke dataset and 72 unique county-level SDH variables from this dataset and the CHR datasets to implement our study objectives. Confirmatory factor analysis (CFA) techniques were applied to create constructs of social determinants of AHM non-adherence.

### 2.1. Data Sources

The CDC Atlas of Heart Disease and Stroke (2014–2016 cycle) and the 2016 CHR datasets were linked by unique five-digit Federal Information Processing Standard (FIPS) codes of each county present in both datasets which are publicly available [22,23]. The CDC Atlas consists of county-level estimates of all heart disease mortality and hospitalizations based on data from the Deaths National Vital Statistics System and Centers for Medicare and Medicaid Services Medicare Provider Analysis and Review (MEDPAR) file, Part A, respectively. This database also contains county-level measurements of risk factors, social and economic factors, health care delivery and insurance and health care costs data derived from multiple data sources. The CHR database is the most comprehensive dataset created specifically to characterize nearly all counties (3000) by health outcomes (length of life and quality of life) and overall health factors (health behaviors, clinical care, social and economic factors, and physical environment) [23].

### 2.2. Measurement of County-Level Non-Adherence to AHM

The CDC Atlas provides medication adherence data calculated from prescription drug claims data for Medicare Advantage or Medicare fee-for-service beneficiaries aged ≥65 years with Medicare Part D coverage. AHM non-adherence was operationally defined as the proportion of days covered (PDC) with blood pressure medication for a period of 365 days. PDC < 80% was considered as non-adherence to AHM. PDC is a validated measure of adherence and persistence to medications, especially among patients with repeated fills [24,25,26]. The AHM PDC measure in the CDC Atlas data combined all individual antihypertensive agents into a single class of AHM—while still a valid measure, this could potentially underestimate non-adherence, since this method requires a user to stay on only a single AHM agent. All variables derived from Medicare, including PDC, were based on claims data of beneficiaries who were ≥65 years old on the extract year. Therefore, for the 2014–2016 CDC Atlas data, non-adherence was measured for the 2015–2016 period. Counties with very small or no Medicare Part D beneficiaries did not have measures of prevalence of adherence in the CDC Atlas datasets since PDCs were calculated from Medicare Part D claims data. Further details for the inclusion of Medicare data for the calculation of non-adherence are available at https://www.cdc.gov/dhdsp/maps/atlas/index.htm [22]. While adherence is an individualized behavior, population-level adherence is an important study outcome because: (1) small changes in population-level adherence could result in larger benefits in population health outcomes such as lower rates of hospitalizations and health care costs [27,28,29]; (2) population-level adherence measures are required for developing interventions geared at improving adherence among groups of patients [27,28]; (3) population-level adherence is increasingly being used as a quality indicator for the performance of providers and individual physicians [30,31].

### 2.3. SDH Variables

The CHR model describes communities with respect to how healthy they are (health outcomes) and existing modifiable factors that predict future health (health factors). For the purposes of ranking counties by these factors, the CHR calculated weighted composite scores for domains of both community characteristics. The health outcomes domains are (1) length of life and (2) quality of life; the health factors domains are (1) health behaviors, (2) clinical care, (3) social and economic factors, and (4) physical environment. The individual variables that make up each domain in the 2016 CHR dataset are time-fixed variables measured at single time points or over an interval that spans 2007–2014 for some variables [32]. The rationale and methods for creating these domains and composite scores have already been published [23]. Prior to the CFA, all variables were standardized to the means of counties within each state using methods previously applied by CHR investigators to create SDH domains [23]. This standardization was necessary for the current analysis because the variables included in our analysis had different measurement scales.

### 2.4. Definition of Social Determinants of Medication Non-Adherence (SDN) Constructs

There are no standardized frameworks for identifying and studying the relationships between SDH constructs and medication non-adherence. Therefore, we used confirmatory factor analysis (CFA) techniques to create a SDN measurement model to relate the observed data to four latent SDN constructs, namely, food insecurity, poverty, weak social support, and healthy built environments (Figure 1). This four-factor CFA measurement model was created on the basis of existing literature on the relationships between contextual SDH factors and medication non-adherence [15,17,19], as well as the availability of variables in the CHR-CDC Atlas linked dataset. We decided to define social determinants of non-adherence constructs rather than use the predefined SDH domains in the CHR dataset for the following reasons: (1) the CHR-based SDH factors included only 30 variables while leaving out nearly 27 other variables that may be important predictors of medication non-adherence; (2) additional potential predictors of medication non-adherence included in the CDC Atlas dataset were not included in defining SDH domains by CHR; (3) the five domains of SDH defined by CHR were based on their relative contributions to health outcomes (length and quality of life), and therefore may not have the same impact on a health behavior such as AHM non-adherence [33].

### 2.5. Statistical Analyses

Statistical analyses were implemented with SAS (version 9.4) and IBM SPSS AMOS (version 25). The goal of these analyses was to describe and quantify the county-level variation in non-adherence to AHM that is explained by social determinants of non-adherence constructs. Structural equation models (SEMs) were used to measure the proportion of variation in county-level AHM non-adherence explained by social determinants of non-adherence constructs identified through the application of PCA and CFA techniques. Briefly, SEM is a hybridized form of analysis of variance (ANOVA), and CFA is well suited to investigating complex relationships between dependent and independent variables. The hypothesized paths and relationships that were tested are represented in Figure 2. Because variables were standardized prior to creating social determinants of non-adherence constructs, there was no need to account for potential clustering at the state level when estimating variance and calculating p-values for the associations between constructs of social determinants of non-adherence and AHM non-adherence.

Only counties with complete data for non-adherence measurements and variables included in the creation of SDH constructs were included in the analysis. The geographic patterns of AHM non-adherence prevalence were described visually with the CDC Atlas maps tool. Although cross-sectional CDC and CHR datasets were used for these analyses, the primary dependent variable was measured in 2015, whereas all predictors were measured in the preceding years, 2012–2014.

## 3. Results

The weighted mean prevalence of AHM non-adherence (PDC < 80%) in 2015 among the 2067 counties (out of 3000) included in the analysis was 25.0%, with a standard deviation (SD) of 18.8%. There was a 7.1% difference in the prevalence of AHM non-adherence between the upper 90th (28.5%) and lower 10th (21.4%) percentiles of counties ranked by the prevalence of AHM non-adherence. A higher prevalence of non-adherence was concentrated within states located in the South (mean = 28.4; SD, 3.2) compared to the Midwest, where the prevalence was the lowest (mean = 20.9; SD, 2.9) (Figure 3).

Table 1 describes the distribution of indicators of the SDN constructs and county-level demographic factors across geographic regions. The proportion of county populations aged ≥65 years were not significantly different across the four geographic regions. The rest of the demographic measures were unequally distributed across geographic regions. More than half of the counties in the South were considered to be rural, while only approximately one-third of the Northeastern counties were rural. The county-level prevalence of all indicators of the SDN constructs differed significantly across geographical regions. The prevalence of all indicators of poverty was highest in the South. While the prevalence of households with single parents and female heads was highest in the South, and the Southern (31.1%) counties were less segregated compared to the Midwestern (35.0%) and Northeastern regions (39.6%). Similarly, there were fewer parks per county (*P* < 0.01), and access to exercise opportunities was lowest in the South (*P* < 0.01). The prevalence of severe housing problems was highest the West (18.6%) compared to a low of 13.0% in the Midwest (*P* < 0.01).

### 3.1. Confirmed Social Determinants of Non-Adherence Model

The hypothesized four-factor CFA model was not confirmed by the data. Model fit was assessed based on the following indices: chi-square, Akaike Information Criterion (AIC), Root Mean Square Error of Approximation (RMSEA), and Comparative Fit Index (CFI). Appendix A provides a summary of the comparative fit between the four-factor base model (Model 1) and a final three-factor model (Model 2), comprised of poverty/food insecurity, weak social support, and healthy built environments constructs. In this three-factor model, the poverty and food insecurity constructs were collapsed into a single construct labelled poverty/food insecurity because of high intercorrelation between these two constructs, further explanation is provided in the next paragraph.

The factor loadings (standardized regression coefficients) and discriminant validity measures from the hypothesized CFA model were evaluated to determine how well the model captured SDN constructs. In the base model, Model 1, an indicator representing counts of pharmacies had a very low factor loading on the healthy built environment (β = −0.06). The food environment index variable also had standardized loadings >1 in the food insecurity construct (β = −1.39). Therefore, Model 1 was modified by excluding these indicators from their respective constructs to create Model 2. While all remaining indicators loaded well (from 0.23 to 0.90) on their respective constructs, the poverty and food insecurity constructs were highly intercorrelated (*R*^2^ = 0.98), suggesting that Model 2 had a poor discriminant validity. For this reason, indicators of the food insecurity and poverty constructs were collapsed into a new construct labelled poverty/food insecurity. Indicators were sequentially loaded onto this new construct and those with the highest factor loadings were retained. The final poverty/food insecurity construct had high internal consistency (Cronbalch-alpha = 0.83) and comprised of percent below poverty line (%), uninsured (%), food stamp/SNAP recipient (%), and food insecurity (%) variables. This newly created construct was then added to the weak social support and healthy built environment constructs retained from Model 2 to create Model 3. This new model, consisting of three constructs—poverty/food insecurity, weak social support and healthy built environment—had a better discriminant validity based on the intercorrelations between poverty and healthy built environment (*R*^2^ = −0.08), weak social support and built environment (*R*^2^ = 0.25) and between poverty and weak social support (*R*^2^ = 0.81). A test of chi-square difference (χ^2^ = 34.49 (degrees of freedom = 6), *P*-value < 0.001) between Models 2 and 3 indicated that Model 3 was a relatively better fit to the data. Therefore, Model 3 was retained for the structural path analysis. The standardized regression coefficients of indicators per each construct in Model 3 are presented in (Appendix A).

### 3.2. Structural Relationships between AHM Non-Adherence and SDN

The SEM results are summarized in Table 2 and Figure 2. The structural equation analysis confirmed that the relationships between AHM non-adherence and SDH are through associative paths involving three SDN constructs which are subdomains of SDH. All three SDN constructs were independently associated with AHM non-adherence; both poverty (β = 0.31, *P*-value < 0.001) and weak social support (β = 0.27, *P*-value < 0.001) were positively associated with AHM non-adherence. On the contrary, healthy built environment was inversely associated with AHM non-adherence (β = −0.10, *P*-value < 0.01). Of all three, poverty/food insecurity had the strongest association with AHM non-adherence. Together, these three social determinants of medication non-adherence explained one-third (*R*^2^ = 30.3%) of the total variation in county-level non-adherence to AHM (Figure 2).

The independence of these AHM non-adherence and SDN relationships from demographic factors were tested in sensitivity analysis. County-level distributions of race (% African-American), gender (% Female), and proportion of rural areas were included in the SEMs as exogenous variables to control for their potential confounding effects. The effects of all three SDN constructs remained statistically significant even after adjusting for these demographic factors, Appendix A. This suggests that the associative relationships between all three SDN constructs and AHM non-adherence are independent of these demographic factors.

## 4. Discussion

Non-adherence to AHM was prevalent nationally and varied by regional and county-level locations. Using a theoretical framework and a SEM approach, poverty/food insecurity, weak social supports and healthy built environments were observed to be associated with AHM non-adherence among Medicare Part D beneficiaries. These findings build on a limited but growing body of literature on the relationships between residential locations and medication non-adherence.

Previous studies on this topic have been focused on testing the relationships between individual-level adherence to oral antidiabetic medications and neighborhood-level social and economic factors among populations with diabetes. Among 749 Mexican-Americans treated for diabetes at a University-affiliated clinic, patients who lived in neighborhoods ranked among the top quartile on neighborhood deprivation index were 60% more likely (odds ratio (OR) = 1.64; 95% CI: 1.12 to 2.39) not to adhere to their medications, compared to those living in the lowest quartile of neighborhoods on the deprivation index [18]. In a much smaller sample (*n* = 179), McClintock et al. (2015) reported that the patients with diabetes who lived in neighborhoods characterized as high in social affluence, residential stability, and neighborhood advantage were approximately eight times (OR = 8.48, 95% CI: 1.71 to 42.02) more likely to adhere to oral antidiabetic medications compared to those living in neighborhoods with fewer of these environmental features [17]. Among participants in the California Health Interview Survey, diabetes patients living in unsafe neighborhoods were more likely to delay filling medications (OR = 1.69, 95% CI: 1.19 to 2.40) [19]. A recent study by Qato et al. (2019) reported a significant reduction in adherence to cardiometabolic medications among patients living in low-income neighborhoods with fewer pharmacies (−7.98%; 95% CI, −8.50% to −7.47%) compared with their counterparts [20]. These data, together with ours, provide evidence that the social, economic and the built environments of residential locations are potentially influential in how populations might adhere to treatment with medications.

There are several implications of our study with respect to advancing research on the topic of social determinants of AHM non-adherence. First, linking pharmaceutical claims data with aggregated databases that contain contextual SDH data is helpful in studying the potential sources of AHM non-adherence beyond individual-level SDH factors. Second, owing to the complexity of large databases that capture several individual contextual SDH variables, dimension reduction methods such as CFA are highly useful in defining SDH constructs. Furthermore, the application of causal frameworks such as SEM can be a great aid in testing specific pathways and relationships between SDH constructs and AHM non-adherence. Third, our findings buttress those from previous studies to show that the social, economic and the built environments are associated with AHM non-adherence. This county-level data could be useful in formulating population-level interventions to improve AHM adherence in patient populations. Individual-level data may be liable to ecological fallacy when applied to address a population-level outcome such as county-level AHM non-adherence [34].

### 4.1. Limitations

This study has limitations that should be considered when interpreting findings. First is ecological fallacy—our findings may not directly translate into patient-level effects because only aggregate-level data was available for analysis. We therefore recommend future research on this topic to investigate the role of contextual SDH factors on AHM non-adherence at the individual level. Second, because only Medicare Part D beneficiaries’ data was used for calculating the prevalence of county-level non-adherence to AHM, nearly one-third of the counties did not have sufficient data on AHM non-adherence due to the very small counts or no Medicare Part D beneficiaries in these counties. Our findings may therefore not be generalizable to counties with predominantly younger (<65 years of age) populations. Third, although PDC is a validated measure of refill adherence it PDC does not reflect primary non-adherence and do not account for gaps in medication refills during hospitalization and out-of-pocket payment for medications [35]. Fourth, although we leveraged aggregate county-level data from the CHR database (one of the most comprehensive databases on SDH) and additional variables from CDC Atlas of Heart Diseases, there were limited measurements of structural SDH factors about policy (social, health and economic) and the built environment. Future studies might well aim to augment the wealth of SDH data in both datasets with additional structural SDH factors from other databases.

### 4.2. Strengths

Our study has some strengths over previous examinations of these issues. First, through the application of dimension reduction methods, we were able to define contextual SDH constructs that are specific to medication non-adherence. Second, the application of SEM allowed us to explore potential determinants of AHM non-adherence at the county level. Third, our findings fill a critical gap in the lack of information on community-level determinants of AHM non-adherence. This information may be important in formulating population-level interventions to improve AHM adherence in communities. Of course, before this data can be adapted for designing interventions, the interactions between contextual and individual-level SDH factors and their cumulative effects on AHM non-adherence must be investigated first.

## 5. Conclusions

AHM non-adherence varies by geographical location, one-third of which is accounted for by contextual SDH factors including poverty/food insecurity, weak social supports and healthy built environments. Given that these SDH constructs accounted for approximately one-third of the variation in AHM non-adherence at the county level, future studies would be well advised to investigate further how these contextual factors interact with individual-level factors to influence AHM non-adherence.

## Figures and Tables

**Figure 1 ijerph-17-06684-f001:**
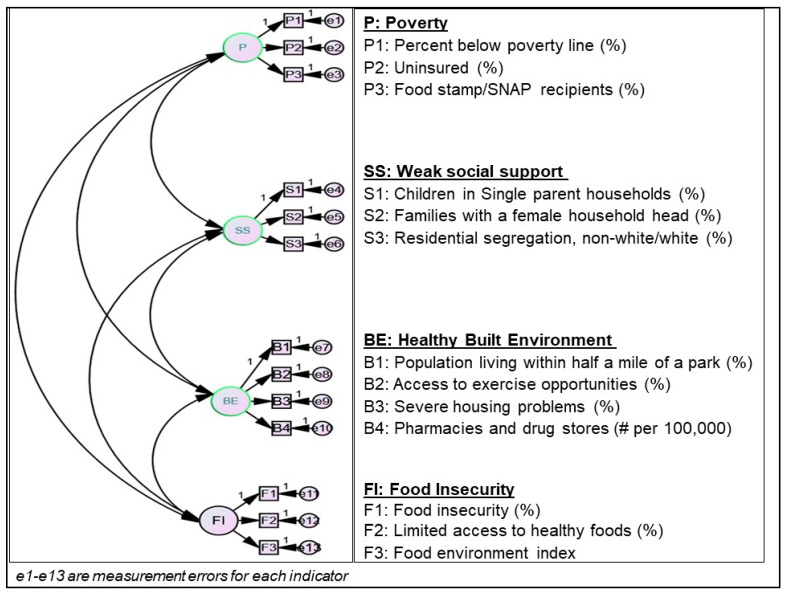
Hypothesized confirmatory factor analysis (CFA) model for measuring social determinants of non-adherence.

**Figure 2 ijerph-17-06684-f002:**
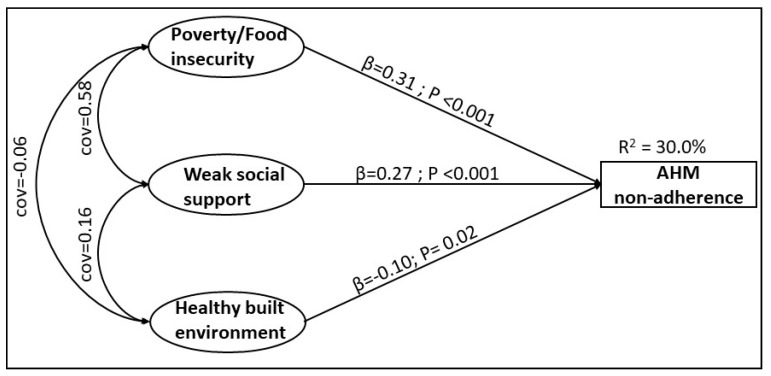
Relationships between social determinants of non-adherence and antihypertensive medication non-adherence. Indicators per construct. Poverty/food insecurity: percent below poverty line (%), uninsured (%), food stamp/SNAP recipient (%), and food insecurity (%). Weak social support: children in single-parent household (%), families with female household head (%), and residential segregation, non-white/white (%). Healthy built environment: population living within half a mile of a park (%), severe housing problems (%), and access to exercise opportunities (%).

**Figure 3 ijerph-17-06684-f003:**
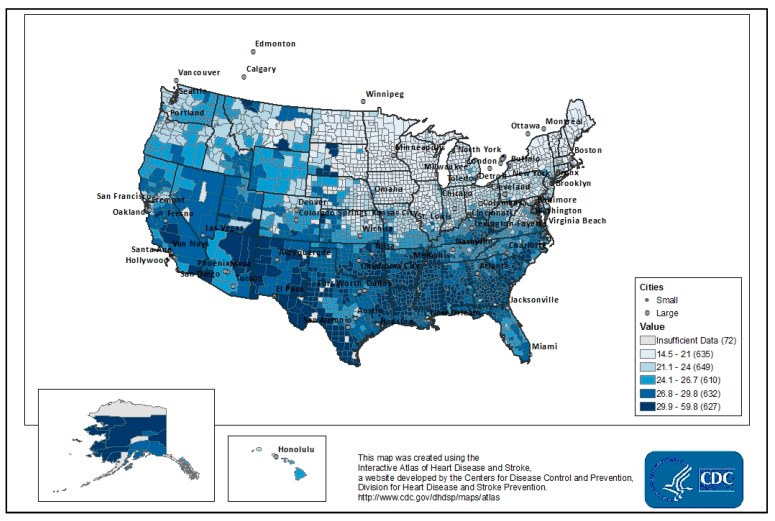
Geographic distribution of the prevalence of non-adherence to antihypertensive medications in the U.S., 2015.

**Table 1 ijerph-17-06684-t001:** Distribution of social determinants of health (SDH) indicators across geographical regions among counties in the U.S., 2015–2016.

Constructs of SDH	Geographic Region, Mean (Standard Deviation)				
	Midwest (*n* = 621)	Northeast (*n* = 197)	South (*n* = 1044)	West (*n* = 205)	*P*-Value
Poverty					
Percent below poverty line (%)	13.2 (4.3)	12.8 (3.8)	18.6 (6.2)	15.7 (5.2)	<0.01
Uninsured (%)	12.9 (3.2)	11.4 (3.1)	19.7 (4.4)	18.9 (3.7)	<0.01
Food stamp/SNAP recipient (%)	12.6 (5.0)	13.0 (4.7)	18.3 (6.8)	15.0 (7.8)	<0.01
Weak Social Network					
Children in single-parent household (%)	29.8 (6.9)	31.5 (7.1)	36.4 (9.3)	31.4 (8.0)	<0.01
Families with female household head (%)	9.7 (2.3)	11.0 (2.8)	13.5 (3.8)	10.7 (3.0)	<0.01
Residential segregation, non-white/white (%)	35.0 (11.9)	39.6 (10.9)	31.1 (12.3)	27.8 (10.3)	<0.01
Built Environment					
Population living within half a mile of a park (%)	27.7 (18.6)	24.8 (20.1)	12.2 (14.4)	32.7 (22.3)	<0.01
Access to exercise opportunities (%)	66.3 (17.1)	76.3 (16.3)	59.9 (22.2)	73.9 (18.0)	<0.01
Severe housing problems (%)	13.0 (3.0)	16.5 (4.7)	15.2 (3.7)	18.6 (5.5)	<0.01
Pharmacies (# per 100,000)	27.7 (18.6)	24.8 (20.1)	12.2 (14.4)	32.7 (22.3)	<0.01
Food insecurity					
Food insecurity (%)	13.3 (2.8)	12.6 (2.2)	16.8 (3.8)	14.8 (2.5)	<0.01
Limited access to healthy foods (%)	5.7 (3.6)	4.2 (2.6)	7.0 (4.8)	8.0 (5.0)	<0.01
Food environment index	7.6 (0.7)	7.9 (0.5)	6.7 (1.1)	7.1 (0.8)	<0.01
Demographic Variables					
Age above 65 (%)	17.1 (3.5)	17.2 (2.7)	16.7 (4.3)	16.6 (5.3)	0.10
African American (%)	3.3 (5.2)	5.2 (6.5)	16.3 (16.6)	1.7 (2.1)	<0.01
Asian American (%)	1.3 (1.6)	2.8 (3.8)	1.3 (1.8)	3.4 (5.4)	<0.01
Pacific Islander (%)	0.1 (0.1)	0.1 (0.1)	0.1 (0.1)	0.3 (0.3)	<0.01
Hispanic (%)	4.4 (4.6)	6.7 (6.1)	9.4 (14.0)	21.3 (18.4)	<0.01
Female (%)	50.0 (1.3)	51.0 (1.3)	50.4 (2.1)	49.7 (1.2)	<0.01
Rural (%)	49.8 (25.1)	42.3 (28.8)	53.1 (28.7)	33.1 (23.3)	<0.01

Abbreviations: SNAP, Supplemental Nutrition Assistance Program.

**Table 2 ijerph-17-06684-t002:** Associations between antihypertensive medication non-adherence and social determinants of non-adherence.

	Model 1 (Unadjusted)		Model 2 (Adjusted)	
	Standardized Regression Coefficient	*P*-Value	Standardized Regression Coefficient	*P*-Value
SDH Constructs				
Poverty/food insecurity	0.31	<0.001	0.38	<0.01
Weak social support	0.27	<0.001	0.12	0.05
Healthy built environments	−0.12	<0.01	−0.13	<0.01
Demographic Factors				
% African American			0.12	<0.01
% Female			−0.02	0.36
% Rural			−0.02	0.36
*R* ^2^	0.30		0.28	

Model 1: Unadjusted. Model 2: Model 1 plus demographic factors. Abbreviations: SNAP, Supplemental Nutrition Assistance Program; *R*^2^, proportion of the variance explained by variables in each model.

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
