# Peer review of "Structural and Social Determinants of Health Factors Associated with County-Level Variation in Non-Adherence to Antihypertensive Medication Treatment"

_ijerph, 2020, doi:10.3390/ijerph17186684_

Round 1
Reviewer 1 Report
I liked the concept behind this study as well as the majority of the methods. Some of you linking between databases was difficult to follow; however your overall design was relatively complex in general. I would also say it took a bit of work to understand what the actual results of the study were because there was some much detail as to how you derived the final model. I would suggest simplifying this section and moving some of this detail to the Discuss section of the paper. I don't think Figure 1 adds to the meaningfulness of your manuscript as you describe this well in the manuscript.
While I think you define this well in your limitations section, the limitation of using on Medicare Part D beneficiary information really is a big limitation. Was there a reason you didn't pursue any Medicaid recipient data?
Overall, this is a good manuscript and I liked the methods by which you derived your model to determine best fit for the analysis.
Author Response
I liked the concept behind this study as well as the majority of the methods. Some of you linking between databases was difficult to follow; however your overall design was relatively complex in general. I would also say it took a bit of work to understand what the actual results of the study were because there was some much detail as to how you derived the final model. I would suggest simplifying this section and moving some of this detail to the Discuss section of the paper. I don't think Figure 1 adds to the meaningfulness of your manuscript as you describe this well in the manuscript.
Response: We sincerely appreciate these comments. We deemed it necessary to describe how we derived our models because these methods are not frequently applied for investigating our topic. We strongly believe that the description of these methods fit better in the methodology section rather than in the discussion section.
While I think you define this well in your limitations section, the limitation of using on Medicare Part D beneficiary information really is a big limitation. Was there a reason you didn't pursue any Medicaid recipient data?
Response: The CDC Atlas of Heart Diseases and Stroke captured only Medicare Part D data. We agree that the inclusion of Medicaid data would have tremendously strengthened the findings of our analysis.
Overall, this is a good manuscript and I liked the methods by which you derived your model to determine best fit for the analysis.

Reviewer 2 Report
Thanks are expressed to the editor and authors for the opportunity to review this interesting paper. Authors comprise an interdisciplinary, multi-institution team with strong, complementary backgrounds. With suitable revision, the methods employed and findings seem to be a good fit (county level variation in social determinants of health for Medicare Part D beneficiaries taking antihypertensive medications) for the International Journal of Environmental Research and Public Health.
Title:
For consideration:
- Slightly change current title -- Structural and social determinants of health factors accounted for county-level variation in non-adherence to antihypertensive medication treatment.
- Replace “accounted for” with “associated with”, thus: Structural and social determinants of health factors associated with county-level variation in non-adherence to antihypertensive medication treatment.
Introduction
- Minor point: Some abbreviations were defined in the Abstract, but not the narrative text. These included, for example: CDC, AHM, CHR.
- More relevant studies addressed some of the social determinants of health for hypertensive patients than cited, at least at the individual level.
Material and Methods:
- Lines 115-118: This reviewer agrees that combining all hypertensive medications in one category may have underestimated adherence. Same is true about use of PDC as measure of adherence. I think use of PDC fine here as was applied to all cases, but maybe note a couple of references that have stated PDC overestimates adherence.)
- Lines 115-118: First clause of that sentence is correctly in the Material and Methods section. The text regarding potential underestimation of non-adherence would be more suitably placed in Limitations section. - Line 140: Abbreviation PCA not previously defined in paper.
Results
- Line 186 (and elsewhere in paper): States SDN but should be SDH.
- Lines 210-211: Conceptually, it was not surprising that food insecurity was not a confirmed factor. From a quick glance, appears food insecurity might have been included in the 4-factor model because references cited (line 150) were mainly from research on use of oral anti-diabetes medications. This could be briefly noted in the Discussion. While there is some correlation, food insecurity for patients with diabetes could be more of an issue than for patients with hypertension. (Many patients have both comorbidities, though that would necessitate a separate analysis, which I am not recommending.)
- Very high correlation of food insecurity with poverty also very understandable. In lines 214-216, and it is good they were combined into a single factor.
- Tables 1 and 2 (starting line 347) contain abbreviations that are not defined in footnotes.
- Table 2 might be slightly misnamed. Factor loadings, per se, were not presented. These are the standardized regression coefficients and p-values.
-For p-values, the word “value” misspelled as “vale” in Table 2 header rows.
Discussion
- Lines 270—272: Recap of results should note as applicable to populations with Medicare Part D coverage.
References
- Reference 22 is just a title.
Author Response
Thanks are expressed to the editor and authors for the opportunity to review this interesting paper. Authors comprise an interdisciplinary, multi-institution team with strong, complementary backgrounds. With suitable revision, the methods employed and findings seem to be a good fit (county level variation in social determinants of health for Medicare Part D beneficiaries taking antihypertensive medications) for the International Journal of Environmental Research and Public Health.
Title:
For consideration:
- Slightly change current title -- Structural and social determinants of health factors accounted for county-level variation in non-adherence to antihypertensive medication treatment.
- Replace “accounted for” with “associated with”, thus: Structural and social determinants of health factors associated with county-level variation in non-adherence to antihypertensive medication treatment.
Response: We have modified the title accordingly.
Introduction
- Minor point: Some abbreviations were defined in the Abstract, but not the narrative text. These included, for example: CDC, AHM, CHR.
Response: We have fixed these in the narrative text.
- More relevant studies addressed some of the social determinants of health for hypertensive patients than cited, at least at the individual level.
Response: We agree that there may be more studies than those cited in this manuscript. However, we limited our background literature review to only those that provide data on the relationships between contextual SDH factors and antihypertensive adherence. As you rightly point out there is ample data on the relationships between individual level SDH and adherence at the individual level but not so much for place-based SDH factors.
Material and Methods:
- Lines 115-118: This reviewer agrees that combining all hypertensive medications in one category may have underestimated adherence. Same is true about use of PDC as measure of adherence. I think use of PDC fine here as was applied to all cases, but maybe note a couple of references that have stated PDC overestimates adherence.)
Response: We agree with your comment about the limitations of PDC as a measure of adherence. We have therefore stated this as a limitation of our findings under the Limitations section as, “Third, although PDC is a validated measure of refill adherence it does not reflect primary nonadherence and do not account for gaps in medication refills during hospitalization and out-of-pocket payment for medications” in lines 370-372.
- Lines 115-118: First clause of that sentence is correctly in the Material and Methods section. The text regarding potential underestimation of non-adherence would be more suitably placed in Limitations section.
Response: We have now moved the following text to the Materials and methods section, “We leveraged data on county-level prevalence of AHM non-adherence from the CDC Atlas of Heart Diseases and Stroke dataset and 72 unique county-level SDH variables from this dataset and the CHR datasets to implement our study objectives. A confirmatory factor analysis techniques (CFA) was applied to create constructs of social determinants of AHM non-adherence.”. See lines 123-126.
- Line 140: Abbreviation PCA not previously defined in paper.
MD: We have deleted PCA since we reported the analysis and results based on only CFA.
Results
- Line 186 (and elsewhere in paper): States SDN but should be SDH.
Response: This typo has been fixed. Thank you.
- Lines 210-211: Conceptually, it was not surprising that food insecurity was not a confirmed factor. From a quick glance, appears food insecurity might have been included in the 4-factor model because references cited (line 150) were mainly from research on use of oral anti-diabetes medications. This could be briefly noted in the Discussion. While there is some correlation, food insecurity for patients with diabetes could be more of an issue than for patients with hypertension. (Many patients have both comorbidities, though that would necessitate a separate analysis, which I am not recommending.)
MD: We appreciate your observation about food insecurity, especially regarding whether food insecurity is associated with antihypertensive adherence as was previously reported for oral anti-diabetes medications. However, there are no published data to support the theory that food insecurity is not a major SDH for adherence to antihypertensive medications and for this reason we do not feel comfortable including this in the discussion.
- Very high correlation of food insecurity with poverty also very understandable. In lines 214-216, and it is good they were combined into a single factor.
- Tables 1 and 2 (starting line 347) contain abbreviations that are not defined in footnotes.
Response: All abbreviations have now been spelt out.
- Table 2 might be slightly misnamed. Factor loadings, per se, were not presented. These are the standardized regression coefficients and p-values.
-For p-values, the word “value” misspelled as “vale” in Table 2 header rows.
Response: Table 2 has been updated to address these concerns.
Discussion
- Lines 270—272: Recap of results should note as applicable to populations with Medicare Part D coverage.
Response: We have made this modification in line 325.
References
- Reference 22 is just a title.
Response: Thank you for pointing this out, we have corrected this error.
